# Parameter Sharing For Heterogeneous Agents in Multi-Agent Reinforcement Learning

## Abstract

Parameter sharing, where each agent independently learns a policy with fully shared parameters between all policies, is a popular baseline method for multi-agent deep reinforcement learning. Unfortunately, since all agents share the same policy network, they cannot learn different policies or tasks. This issue has been circumvented experimentally by adding an agent-specific indicator signal to observations, which we term "agent indication." Agent indication is limited, however, in that without modification it does not allow parameter sharing to be applied to environments where the action spaces and/or observation spaces are heterogeneous. This work formalizes the notion of agent indication and proves that it enables convergence to optimal policies for the first time. Next, we formally introduce methods to extend parameter sharing to learning in heterogeneous observation and action spaces, and prove that these methods allow for convergence to optimal policies. Finally, we experimentally confirm that the methods we introduce function empirically, and conduct a wide array of experiments studying the empirical efficacy of many different agent indication schemes for graphical observation spaces.

## 1 Introduction

Reinforcement Learning (RL) is the intersection of machine learning and optimal control. It allows an agent in an environment to learn a policy which output actions given observations in order to maximize a reward signal defined by the environment. However, many real-life scenarios are more accurately modeled by multi-agent reinforcement learning (MARL), where multiple agents act and learn simultaneously.

MARL methods are often further subdivided according to two main properties: whether the agents are cooperative or competitive (or mixed), and if various components are centralized or decentralized. In decentralized systems, each agent takes decisions and learns independently, without access to other agents' observations, actions, or policies. This is similar to real-life scenarios like groups of animals, as opposed to a 'hive-mind' controller acting as a single large RL agent (the centralized case). However, decentralized learning is intuitively more difficult, as information must be independently learned by each agent or passed between them. Therefore, most modern MARL research follows the paradigm of Centralized Training, Decentralized Execution (CTDE) (Lowe et al., 2017), where agents execute their individual policies separately, but have access to other agents' observations during training. Examples of CTDE include MADDPG (Lowe et al., 2017), COMA (Foerster et al., 2018), and QMIX (Rashid et al., 2018).

This paper specifically considers parameter sharing (which we refer to as "parameter sharing"), a CTDE execution scheme where all policies are represented by a single neural network with the same shared parameters for each. This was introduced by Tan (1993) for classical RL and later introduced to cooperative multi-agent *deep* reinforcement learning by Foerster et al. (2016); Gupta et al. (2017); Chu & Ye (2017). This has gone on to be used in numerous interesting applications However, parameter sharing as been used in numerous interesting applications (Zheng et al., 2018; Yu et al., 2022; Chen et al., 2021; Yang et al., 2018).

Because parameter sharing only has one policy network or function, it is easy to assume that it can only handle agents with identical behavior in the environment, which is a significant limitation. This assumption has been relaxed for different types of agents, by adding an indication of the observing agent to the observations,

allowing a single policy to serve multiple agents (Foerster et al., 2016; Gupta et al., 2017). While this is known to work empirically, our work's first contribution is formally defining "agent indication" and proving that it allows for representing to optimal policies (subsection 3.2).

An additional limitation of parameter sharing even with agent indication is that agents cannot have different action and observation spaces with a neural network that can only accept and produce fixed sized tensors. This work's second contribution is formally introducing two padding based methods that are capable of resolving this for the first time, and similarly proving that these methods allow for convergence to optimal policies (subsection 3.2, subsection 3.3). We additionally provide experimental validation that these methods function in section 4.

One implementation detail that comes up when using agent indication is the specific method you indicate an agent in the observation space with. There are two "natural" methods to do this with vector observation spaces that have been used in the literature, however no literature exists applying this to graphical observation spaces. This work poses five simple methods for agent indication in graphical observation spaces, and conducts experiments based on hyperparameter tuning sweeps to roughly determine the relative performance of these methods, and in doing so gain initial insight into the likely sensitivity environments have to these methods and if any of these methods may be more universally useful than the others.

These experiments additionally experimentally confirm that the methods we theoretically describe function experimentally, as one environment we ran tests on required applying every method we describe in order to be able to learn.

## 2 Background and Related Work

### 2.1 Partially-Observable Stochastic Games

In multi-agent environments, MDPs can be extended to include a set of actions for each agent to create the Multi-agent MDP ("MMDP") model (Boutilier, 1996). However, this model assumes all agents receive the same rewards. The Stochastic Games model (sometimes called *Markov Games*), introduced by (Shapley, 1953), extends this by allowing a unique reward function for each agent. The Partially-Observable Stochastic Games ("POSG") model, defined below, extends the Stochastic Games model to settings where the state is only partially observable (akin to a POMDP); it can be seen as equivalent to a kind of extensive game (Kuhn, 1953). This is a more general model, and what we use in this paper. We define this model for our later use in Definition 1. A good in depth overview of the POSG model and intuition can be found in Terry et al. (2020a).

**Definition 1** (Partially-Observable Stochastic Game). A *Partially-Observable Stochastic Game* (POSG) is a tuple $\langle \mathcal{S}, N, \{\mathcal{A}_i\}, P, \{R_i\}, \{\Omega_i\}, \{O_i\}\rangle$, where:

- $\mathcal{S}$ is the set of possible *states*.

- $N$ is the *number of agents*. The *set of agents* is $[N]$.

- $\mathcal{A}_i$ is the set of possible *actions* for agent $i$. We denote by $U = \prod_{i \in [N]} \mathcal{A}_i$ the set of possible *joint actions* over all agents.

- $P \colon \mathcal{S} \times \prod_{i \in [N]} \mathcal{A}_i \times \mathcal{S} \to [0, 1]$ is the (stochastic) *transition function*.

- $R_i \colon \mathcal{S} \times \prod_{i \in [N]} \mathcal{A}_i \times \mathcal{S} \to \mathbb{R}$ is the *reward function* for agent $i$.

- $\Omega_i$ is the set of possible *observations* for agent $i$.

- $O_i \colon \mathcal{S} \times \mathcal{A}_i \times \Omega_i \to [0, 1]$ is the *observation function*.

### 2.2 Parameter Sharing

Full parameter sharing (which we refer to as "parameter sharing"), where all policies are represented by a single neural network with the same shared parameters was first introduced by Tan (1993) for classical RL

and later concurrently introduced to cooperative multi-agent *deep* reinforcement learning by Foerster et al. (2016); Gupta et al. (2017); Chu & Ye (2017). This simple method has since been used to remarkable efficacy in various applications, such as Zheng et al. (2018); Yu et al. (2022); Chen et al. (2021). It is also worth noting that parameter sharing is equivalent to naive self-play in competitive environments.

### 2.3 Coping with Heterogeneity

Agent indication for parameter sharing was first implemented in Foerster et al. (2016), and has been used in numerous derivative works. We are aware of no work studying the best method of this in graphical observations, theoretically studying methods for this, or attempting to cope with heterogeneous action or observation spaces outside of padding the action space of medivac unit in Samvelyan et al. (2019). In the case of the medivac, actions greater than 5 specify a target allied agent index to heal, as opposed to combat agents, where actions greater than 5 specify an enemy agent index to attack. This overloading of the same range of the action space to do different things to different sets of agents is effectively equivalent to action padding. However, the authors do not justify this choice of action encoding or comment on it in their paper.

## 3 Theoretical Results

In this section, we call the agents of a POSG *homogeneous* if they can be reordered into any other order without changing the behavior of the POSG. Formally we chose to define this as meaning all agents have identical action spaces, observation spaces, and reward functions, and that the transition function is symmetric with respect to permutations on the actions input. If a POSG is not homogeneous, we definite it to be heterogeneous.

Parameter sharing has been traditionally seen as a technique that can only be used in games with homogeneous agents because of the fact that a single neural network is learned and shared among all agents. However, in this section we present a novel method of modifying the observation spaces, and prove that it can allow for the use of parameter sharing in the case of heterogeneous agents.

### 3.1 Disjoint Observation Spaces Allows for Learning an Optimal Policy

When agents are not homogeneous, it is not clear that parameter sharing can be applied. However, we note that a single policy can be used in cases where the observation spaces of the agents are disjoint. We state this claim and prove it, as a preliminary for the full "agent indication" technique described in the following section.

**Lemma 1.** *If $G = \langle \mathcal{S}, N, \{\mathcal{A}_i\}, P, \{R_i\}, \{\Omega_i\}, \{O_i\} \rangle$ is a POSG such that $\{\Omega_i\}_{i \in [N]}$ is disjoint (i.e., $\Omega_i \cap \Omega_j = \emptyset$ for all $i \neq j$), then any collection of policies $\{\pi_i\}_{i \in [N]}$ can be expressed as a single policy $\pi^{[N]} \colon \left( \bigcup_{i \in [N]} \Omega_i \right) \times \left( \bigcup_{i \in [N]} \mathcal{A}_i \right) \to [0, 1]$ which, from the perspective of any single agent $i$, specifies a policy equivalent to $\pi_i$.*[1]

*Proof.* Let $\Omega = \bigcup_{i \in [N]} \Omega_i$ be the set of all observations across all agents, and similarly define $\mathcal{A} = \bigcup_{i \in [N]} \mathcal{A}_i$ to be the set of all actions available to agents. Note that while $\Omega$ is a union of $N$ disjoint sets, it is not necessarily true that $\{\mathcal{A}_i\}_{i \in [N]}$ is disjoint and so $\mathcal{A}$ is not necessarily the union of disjoint sets.

Define $\iota \colon \Omega \to [N]$ as follows: $\iota(\omega)$ is the (unique) agent $i$ for which $\omega \in \Omega_i$. Thus, for all $\omega \in \Omega$, we have that $\omega \in \Omega_{\iota(\omega)}$. Note that $\iota$ is well-defined specifically because the observation sets are disjoint, and thus each observation $\omega \in \Omega$ appears in exactly one agent's observation space.

Now, we define our single policy $\pi^{[N]} \colon \Omega \times \mathcal{A} \to [0, 1]$. Let

$$\pi^{[N]}(\omega, a) = \begin{cases} \pi_{\iota(\omega)}(\omega, a) & \text{if } a \in \mathcal{A}_{\iota(\omega)} \\ 0 & \text{otherwise} \end{cases} \tag{1}$$

---

[1] Formally, for any agent $i \in [N]$, observation $\omega \in \Omega_i$, and action $a \in \mathcal{A}_i$, $\pi^{[N]}(\omega, a) = \pi_i(\omega, a)$.

One can see from this definition that for any agent $i \in [N]$, for any $\omega \in \Omega_i$, and for any $a \in \mathcal{A}_i$, we have $\pi^{[N]}(\omega, a) = \pi_i(\omega, a)$. Thus, from the view of agent $i$, $\pi^{[N]}$ defines a policy consistent with its own policy $\pi_i$. $\qquad\square$

**Corollary 1.** *For any Partially-Observable Stochastic Game $G = \langle \mathcal{S}, N, \{\mathcal{A}_i\}, P, \{R_i\}, \{\Omega_i\}, \{O_i\} \rangle$ with disjoint observation spaces, there exists a single policy $\pi^* \colon (\bigcup_{i \in [N]} \Omega_i) \times (\bigcup_{i \in [N]} \mathcal{A}_i) \to [0, 1]$ which is optimal for all agents; i.e. $\forall i \in [N], \omega \in \Omega_i, a \in \mathcal{A}_i$, we have $\pi^*(\omega, a) = \pi_i^*(\omega, a)$, where $\pi_i^*$ is an optimal individual policy for agent $i$.*

To briefly recap the intuition of the method we propose in this section—if by any means, the identity of an agent is indicated in the observation space passed to a full parameter sharing based learning method during training, then the policy will be able to learn to distinguish each agent and act in a manner adapted to that agent specifically, despite only being a single network.

### 3.2 Agent Indication Allows for Representing Optimal Policies

When observation spaces are not disjoint—that is, the raw observations from the environment do not indicate the identity of the agent—we can force them to be disjoint by "tagging" the observations with an identifier unique to each agent. This technique is referred to as "agent indication" and is described formally below.

**Theorem 1.** *For every POSG, there is an equivalent POSG with disjoint observation spaces.*

*Proof.* Let $G = \langle \mathcal{S}, N, \{\mathcal{A}_i\}, P, \{R_i\}, \{\Omega_i\}, \{O_i\} \rangle$ be a POSG with non-disjoint observation spaces. We define $G' = \langle \mathcal{S}, N, \{\mathcal{A}_i\}, P, \{R_i\}, \{\Omega_i'\}, \{O_i'\} \rangle$, where $\Omega_i'$ and $O_i'$ are derived from $\Omega_i$ and $O_i$ respectively, as described below.

For each agent $i$, we define $\Omega_i' = \Omega_i \times \{i\} = \{(\omega, i) \mid \omega \in \Omega_i\}$. Intuitively, we "attach" information about the agent $i$ to the observation. Now, for each agent $i \in [N]$, we define $O_i' \colon \mathcal{A}_i \times \mathcal{S} \times \Omega_i' \to [0, 1]$ as $O_i'(a, s, (\omega, i)) = O_i(a, s, \omega)$. This is equivalent to $G$ in the sense that there is a family of bijections $f_i \colon \Omega_i \to \Omega_i'$ such that $\forall i \in [N], \forall a \in \mathcal{A}_i, \forall s \in \mathcal{S}, \forall \omega \in \Omega_i, O_i(a, s, \omega) = O_i'(a, s, f_i(\omega))$ (specifically, $f_i(\omega) = (\omega, i)$). $\qquad\square$

Theorem 1 together with Corollary 1 shows that an optimal single policy $\pi^* \colon (\bigsqcup_{i \in [N]} \Omega_i) \times (\bigcup_{i \in [N]} \mathcal{A}_i) \to [0, 1]^2$ is consistent with the optimal policies of each agent: if $\pi_i^* \colon \Omega_i \times \mathcal{A}_i \to [0, 1]$ is an optimal individual policy for agent $i$, then $\pi^*((\omega, i), a) = \pi_i^*(\omega, a)$ for every action $a \in \mathcal{A}_i$.

In practice, many environments lend themselves to disjoint observation spaces (e.g., games with a third-person point of view), and Lemma 1 shows that one can use a single algorithm (e.g., a single neural network) to learn a single policy that behaves differently for each agent. Additionally, Theorem 1 says that for non-disjoint observation spaces, we can "attach" the identity of each agent to its observations (e.g. by superimposing an identifier that is distinct for each agent onto the observations before they are input to the learning algorithm), forcing the modified observations to be elements of disjoint observation spaces so that Lemma 1 applies. Corollary 1 says that there is no disadvantage to this approach in terms of the optimality of the learned policy. In situations where the representations of the observations of each agent do not have the same size, they can all be "padded" to the size of the largest and learning can proceed as normal. In other words, if you add 0s (or a similar value) to pad observation tensors with a smaller shape than others such that they are the same size, then this can be passed to a policy network and it can learn as normal. This padding allows neural networks or other policies of fixed dimension input to control agents with differing observation spaces.

### 3.3 Padding Heterogeneous Action Spaces Allows for Representing Optimal Policies

Consider an environment where each agent $i \in [N]$ has a different action space $\mathcal{A}_i$ and, critically, that for some pair of agents $i, i' \in [N]$ we may have $|\mathcal{A}_i| \neq |\mathcal{A}_{i'}|$. In the formal model, this is not a problem, but it does present a minor issue in implementation. Specifically, suppose we have a learning algorithm that has learned a policy $\pi \colon \omega \times \mathcal{A} \to [0, 1]$ for a single agent. In practice, the learning algorithm, given an observation $\omega$, outputs the induced probability distribution $\pi(\omega, \cdot)$ often as a vector $\vec{a} \in [0, 1]^{|\mathcal{A}|}$ with $\ell_1$-norm 1.

---

[2] $\bigsqcup_i \Omega_i$ is the *disjoint union*: $\bigsqcup_{i \in [N]} \Omega_i := \bigcup_{i \in [N]} (\Omega_i \times \{i\}) = \bigcup_{i \in [N]} \{(\omega, i) \mid \omega \in \Omega_i\}$.

If each agent's behavior is learned as a separate policy, the learning algorithm for agent $i$ will output a probability vector in $[0,1]^{|\mathcal{A}_i|}$. However, if there are two agents $i, i'$ with $|\mathcal{A}_i| \neq |\mathcal{A}_{i'}|$, using the same network for both agents $i$ and $i'$ appears to preclude the use of parameter sharing, as the output vectors have different dimensions.

We can address this as follows. Suppose we wish for our algorithm to learn a single policy $\pi^{[N]}$ for all agents, as in the previous section. One option is to have the algorithm output a probability vector in $[0,1]^{|\mathcal{A}|}$, where $\mathcal{A} = \bigcup_{i \in [N]} \mathcal{A}_i$ as before. Now, when the algorithm outputs a vector $\vec{a} \in [0,1]^{\mathcal{A}}$ for agent $i$, we can simply "clip" the vector and consider only the subvector corresponding to actions in $\mathcal{A}_i$. We can write this formally as $\vec{a}_{\mathcal{A}_i} \in [0,1]^{\mathcal{A}_i}$.

This can be quite space-inefficient, though, as in the worst case (when all agents have disjoint action spaces) $|\mathcal{A}| = \sum_{i \in [N]} \mathcal{A}_i$. Further, since an agent $i$ will never perform an action $a \notin \mathcal{A}_i$, much space is wasted representing the output as a sparse vector (i.e. one with many zeros). A more practical alternative is to instead output a vector whose dimension is only as large as the largest action space, padding the vector with zeros for agents with smaller action spaces. Formally, the learning algorithm can simply output a vector $\vec{a} \in [0,1]^{\alpha}$ for all agents, padding zeros at the end where necessary. An agent $i$ with $|\mathcal{A}_i| < \alpha$ receiving a vector $\vec{a} \in [0,1]^{\alpha}$ can simply consider the subvector $\langle \vec{a}_1, \vec{a}_2, \ldots, \vec{a}_{|\mathcal{A}_i|} \rangle$. Essentially, the learning algorithm pads the action vector to length $\alpha$, and each agent clips the vector they receive to length $|\mathcal{A}_i|$.

To briefly recap the intuition of the method we propose in this section—if you can "pad" action spaces so that a neural network is outputting actions of a fixed dimensionality and range, agents that can only accept less than this can either clip or throw out unneeded actions and the policy can still be represented. This padding allows neural networks or other policies of fixed dimension output to control agents with differing action spaces.

## 4 Experimental Results

### 4.1 Overview

One important implementation detail in using agent indication is the method by which you indicate the agent in the observations. In the case of 1D vector observations, previous works have done this via one hot or binary encoding, depending on the type of environment. However in the case of graphical observation spaces, which are very common, there are an incredibly large number of possible ways to indicate an agent, and no prior works experimentally investigate the impact of these on the end performance of typical deep reinforcement learning regimes, which could be profound.

To this end, this section proposes five reasonable simple approaches to agent indication for graphical observations (there are no past works to draw from), specifically trying to pick the simplest approaches we were able to think of. To evaluate the relative success of these approaches, we selected five interesting graphical environments (subsection 4.3) and conducted large hyperparameter searches to find the best parameter shared PPO policy over each where the agent indication method was included as one hyperparameter. Each hyperparameter search included all the commonly searchers hyperparameters for PPO, detailed in Appendix B. We then retrained the best set of 10 hyperparameters 10 times each for each environment (5 for prospector), and present the expected reward of the policy during training averaged across all of the training runs.

Additionally, learning in the *prospector* environment required using both agent indication and the two methods previously introduced for coping with action space and observation space heterogeneity, showing that the methods are able to empirically learn.

We caution that determining a universally best agent indication method is likely not possible and the success of any method will depend on various details in the environment and training regime. However, these experiments can give insight regarding how much the different methods may matter in practice and if a "generally best" agent indication method of those we postulate exists or if it tends to be very environment specific. This experimental design was chosen in order to mitigate the likely profound confounding impact other hyperparameters may have on the performance of a specific agent indication method.

### 4.2 Agent Indication Methods Tested

Per the reasoning described in the previous section, this work seeks to gain insight into the relative performance of the following five agent indication methods:

1. *Identity* — Do nothing to the observation, included for control.

2. *Geometric* — Add an additional channel with a per-pixel checkerboard for one type of agent and checkerboard translated by one pixel for the other type.

3. *Binary* — Add additional channels, each of which is entirely black or white based on the type of an agent.

4. *Inversion* — Invert the color of the observation of certain types of agents and add it as a channel to the original observation. For agents that do not need inversion, duplicate original observation.

5. *Inversion with Replacement* — The same as *Inversion*, but the inverted observation (or the same duplicate observation for agents that do not need it) is used in place of the original observation.

### 4.3 Environment Selection

We selected five environments from this work, all from PettingZoo (Terry et al., 2020a), a library with over 60 diverse multi-agent reinforcement learning environments with a standardized API. PettingZoo has two "classes" of environments with graphical observations- the "Butterfly" class with custom made games with pygame rendering and pymunk physics with a diverse set of rules, as well as the multi-player Atari games introduced in Terry et al. (2020c). To maximize environment diversity, we chose every environment in the Butterfly class with heterogeneous agents (three), and two Atari games (the choice of which we will explain below).

The environments we chose were:

- *Cooperative Pong* — A Butterfly environment inspired by the Atari pong environment, where two differently shaped pistons work together to keep the ball in the air for as long as possible.

- *Knights Archers Zombies* — A Butterfly environment where knight and archer agents collectively work together to prevent zombies from crossing the bottom of the screen.

- *Prospector* — A Butterfly environment where banker and prospector agents work together to collect gold, give to one another and deposit it in banks

- *Pong* — classic two player Atari pong

- *Entombed Cooperative* — an Atari game where two identical agents work together to progress to the bottom of the screen

More comprehensive documentation of all these environments is available at `pettingzoo.ml`.

All of the butterfly class environments are cooperative environments (which is when parameter sharing is conventionally described as being used), with heterogeneous types of agents. Entombed cooperative was chosen because it is the only cooperative multiplayer Atari game supported in PettingZoo (few were ever made), and Pong was chosen as a simple test of the impact of agent indication in a simple and very well known competitive environment. Note that the Atari environments have identical baseline observations for each agent (the pixels on the screen), making agent indication crucial to allow the policy to differentiate at all between the agents. All environments are pictured in Figure 2.

### 4.4 Additional Implementation Details

Prior to agent indication, each environment had the following preprocessing done to its observations using SuperSuit (Terry et al., 2020b).

1. RGB images converted to grayscale using `color_reduction_v0`

2. Image resized to 96x96 pixels via bi-linear interpolation with `resize_v0`.

3. Stacked the last 4 frames along the channel dimension with `frame_stack_v1`.

4. In Knights Archers Zombies only, since individual agents can terminate early in this environment, and our training setup did not support this explicitly, we used `black_death_v2` to create an all black image to represent the terminal state for the dead agent.

The following number of training timesteps for each environment were used:

- *Cooperative Pong* — 4 million timesteps.

- *Knights Archers Zombies* — 10 million timesteps.

- *Prospector* — 100 million timesteps.

- *Pong* — 10 million timesteps.

- *Entombed Cooperative* — 10 million timesteps.

Additionally, the hyperparameter ranges we searched are described in Appendix B, and for the evaluation in the Pong environment, we evaluated our agent's performance by letting it play against the built-in bot implemented in the game's single player mode (as the rewards when playing against itself were not meaningful). All of our learning code was implemented by Stable Baselines 3 (Raffin et al., 2021), and our hyperparameter search code was based upon RL Baselines3 Zoo (Raffin, 2020). All code used in our experiments is available at `https://anonymous.4open.science/r/parameter-sharing-paper-EB2D/`.

### 4.5 Results

The 10 best hyperpameters for each environment found during each automated hyperparameter search were all retrained 10 times (except for 5 in the case of prospector due to the longer run time). For each training process, the average-per-agent reward of the policy checkpoint were taken and averaged, in order to ensure that the rewards associated with hyperparameters are accurate and representative. These averaged-per-agent rewards are displayed in Figure 1 and are detailed in Appendix A.

These results show a that environments appear to be highly sensitive to agent indication method given the variance and specificity of which methods worked will in different environments. The exception to this is that inversion held a place in the set of best hyperparameters for every environment and was *the* best for three out of five environments. This likely means that inversion is a good initial starting point for practitioners.

Additionally, as Prospector, an environment with heterogeneous action and observation spaces and two types agents, was able to be learned with parameter sharing using agent indication and the two padding methods this work introduces, this shows that these methods we introduce can empirically allow for learning.

## 5 Conclusion

This paper introduces novel methods for coping with heterogeneous action and observation spaces in multi-agent environments learned via full parameter sharing, and shows that these and the previously known method of agent indication (for coping with heterogeneous agent behaviors) are able to represent optimal policies. We show that these methods are also capable of working together to empirically allow for learning.

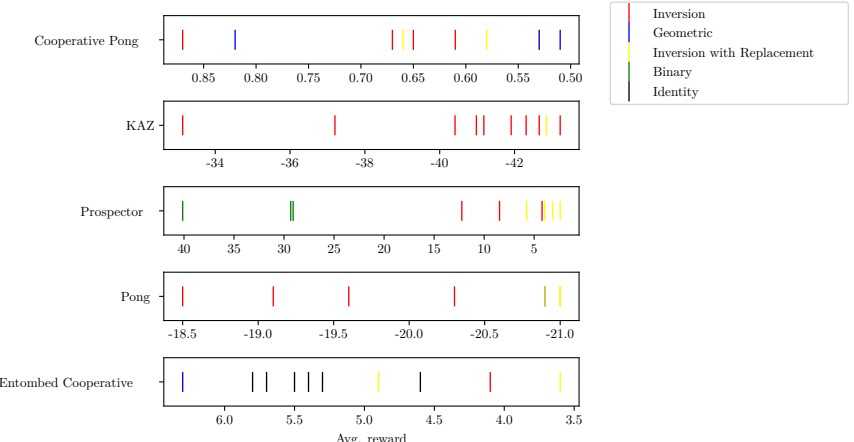

Figure 1: Average reward over 10 training run (with different seeds) of the 10 best hyperparameter/agent indication combinations

We further offer experiments on the efficacy of different methods of agent indication in graphical observation spaces, a previously unstudied problem, showing that it appears to be highly environment dependent, but that inverting the observations associated with one agent is likely a good baseline and starting point for researchers approaching new environments.

We hope these results will enable greater practical application of parameter sharing for multi-agent reinforcement learning.

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

# A    Reward and Agent Indication Method Tables

Table 1: 10 best agent indication methods for Cooperative Pong. 10 additional evaluations were done for each of the hyperparameters.

| Agent indication method | Avg. Reward |
|---|---|
| Inversion | -33.13 |
| Inversion | -37.20 |
| Inversion | -40.41 |
| Inversion | -40.98 |
| Inversion | -41.18 |
| Inversion | -41.91 |
| Inversion | -42.31 |
| Inversion | -42.66 |
| Inversion with Replacement | -42.85 |
| Inversion | -43.22 |

Table 2: 10 best agent indication methods for KAZ. 10 additional evaluations were done for each of the hyperparameters.

| Agent indication method | Avg. Reward |
|---|---|
| Inversion | 0.87 |
| Geometric | 0.82 |
| Inversion | 0.67 |
| Inversion with Replacement | 0.66 |
| Inversion | 0.65 |
| Inversion | 0.61 |
| Inversion with Replacement | 0.58 |
| Inversion with Replacement | 0.53 |
| Geometric | 0.53 |
| Geometric | 0.51 |

Table 3: 10 best agent indication methods for Prospector. 5 additional evaluations were done for each of the hyperparameters.

| Agent indication method | Avg. Reward |
|---|---|
| Binary | 40.13 |
| Binary | 29.33 |
| Binary | 29.09 |
| Inversion | 12.23 |
| Inversion | 8.46 |
| Inversion with Replacement | 5.75 |
| Inversion | 4.21 |
| Inversion with Replacement | 3.95 |
| Inversion with Replacement | 3.14 |
| Inversion with Replacement | 2.40 |

Table 4: 10 best agent indication methods for Pong. 10 additional evaluations were done for each of the hyperparameters.

| Agent indication method | Avg. Reward |
|---|---|
| Inversion | -18.5 |
| Inversion | -19.1 |
| Inversion | -19.6 |
| Inversion | -20.3 |
| Geometric | -20.9 |
| Inversion with Replacement | -20.9 |
| Inversion with Replacement | -21.0 |
| Inversion with Replacement | -21.0 |
| Inversion with Replacement | -21.0 |
| Inversion with Replacement | -21.0 |

Table 5: 10 best agent indication methods for Entombed Cooperative. 10 additional evaluations were done for each of the hyperparameters.

| Agent indication method | Avg. Reward |
|---|---|
| Geometric | 6.3 |
| Identity | 5.8 |
| Identity | 5.7 |
| Identity | 5.5 |
| Identity | 5.4 |
| Identity | 5.3 |
| Inversion with Replacement | 4.9 |
| Identity | 4.6 |
| Inversion | 4.1 |
| Inversion with Replacement | 3.6 |

# B Hyperparameter Search Range

## B.1 PPO

Table 6: Hyperparameter search range for PPO algorithm. Continuous range is noted in square brackets.

| HYPERPARAMETER | RANGE |
|---|---|
| BATCH SIZE | 8, 16, 32, 64, 128, 256, 512 |
| NUMBER OF TIMESTEPS PER UPDATE | 8, 16, 32, 64, 128, 256, 512, 1024, 2048 |
| DISCOUNT FACTOR (GAMMA) | 0.9, 0.95, 0.98, 0.99, 0.995, 0.999, 0.9999 |
| ENTROPY COEFFICIENT | [0.00000001, 0.1] |
| CLIP RANGE | 0.1, 0.2, 0.3, 0.4 |
| NUMBER OF EPOCHS | 1, 5, 10, 20 |
| GAE COEFFICIENT (LAMBDA) | 0.8, 0.9, 0.92, 0.95, 0.98, 0.99, 1.0 |
| MAXIMUM GRADIENT NORM | 0.3, 0.5, 0.6, 0.7, 0.8, 0.9, 1, 2, 5 |
| VALUE FUNCTION COEFFICIENT | [0, 1] |
| NETWORK ARCHITECTURE | (64, 64), (256, 256) |
| ACTIVATION FUNCTION | TANH, RELU |

## C Environment Images

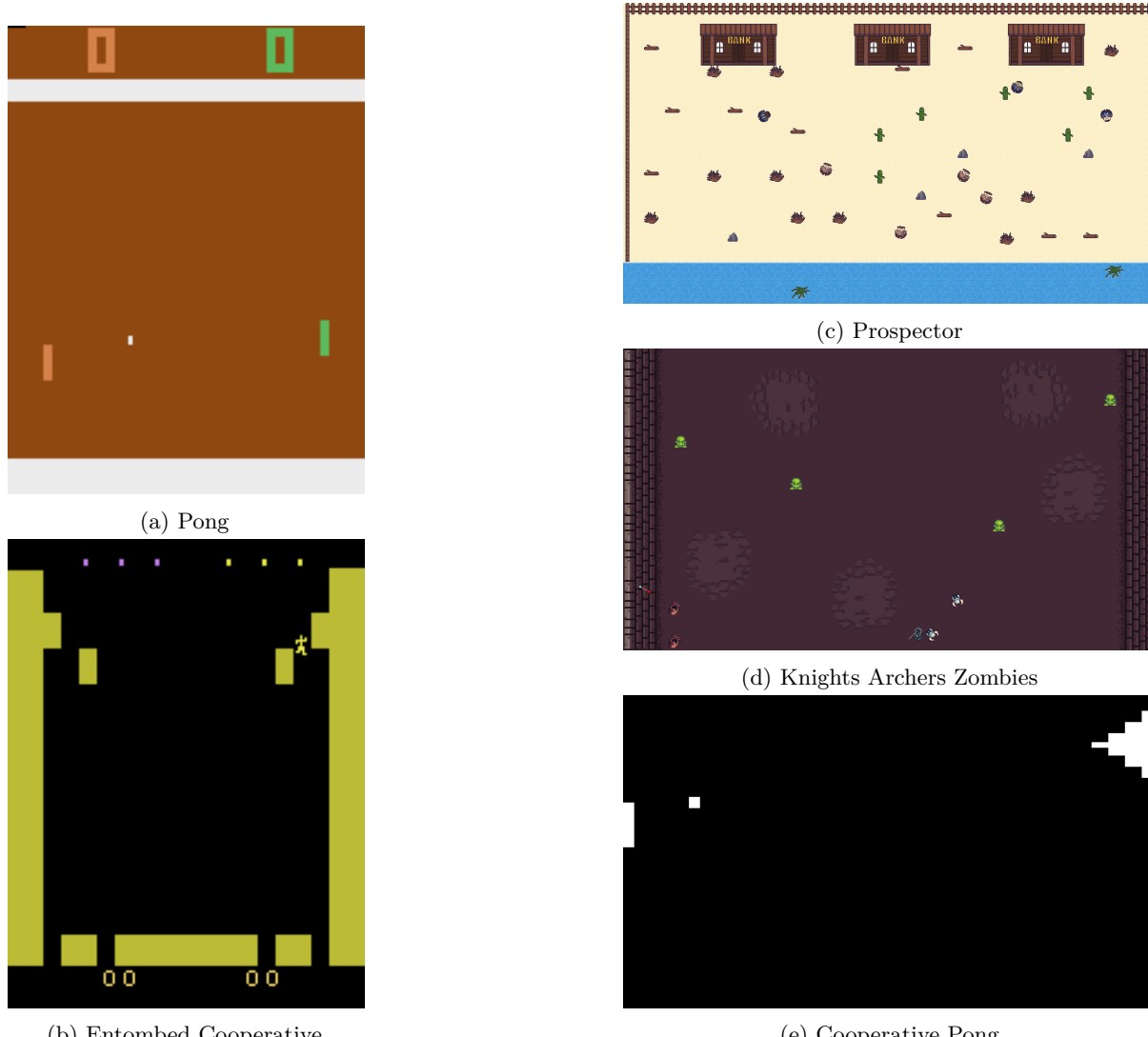

(a) Pong

(b) Entombed Cooperative

(c) Prospector

(d) Knights Archers Zombies

(e) Cooperative Pong

Figure 2: Images of the benchmark environments from Terry et al. (2020a).

