# OpenReview forum: "Parameter Sharing For Heterogeneous Agents in Multi-Agent Reinforcement Learning"
_TMLR — Rejected by TMLR_

### Review · Reviewer_s7mr · 2022-04-29

**Summary Of Contributions:**

In this paper the authors investigate the role of 'agent indicators' when using a single policy to represent and learn the policies of multiple distinct agents in a multi-agent setting. Specifically, on how it can allow for different agents to learn distinct policies despite sharing a single function to compute their policy. The paper first demonstrates that this is not required in the situation where the observations of each agent are disjoint, and then shows that by using an appropriate agent indicator (the agent id in this case) any set of observations can be made disjoint. The paper also presents a scheme for enabling a single policy to be used for all agents even if they have action spaces of varying dimensions through the use of padding. Finally, the paper conducts experiments comparing a set of indicator methods designed for use with image-based observations.

**Broader Impact Concerns:**

No concerns.

**Requested Changes:**

# Critical

In its current form I cannot recommend the paper for acceptance. I do not think that the theoretical results are substantial enough to warrant publication since they are already well understood in the literature (they might not have been explicitly written down as a lemma or theorem, but they are certainly known).

One area of this paper that would be of interested to the community is the experimental results on agent indicators, but they would need to be substantially improved:
First, experimental results on non-image observation spaces since the method of agent indication is as much of an open question there as it is with images (at the very least a *good* justification for why only image-based envs are considered).
Second, the analysis, discussion and presentation of the results should be significantly expanded upon (there are some suggestions in the previous section under weaknesses).

Additionally, you should provide the actual hyperparameters that you found were the best for each method, as well as a graph of some kind to show the full distribution of results for each method as well as full learning curves to show sample efficiency. You seem to have run a lot of experiments, but the presentation and discussion of their results is lacking at the moment.

In Lemma 1 your characterisation of disjointness is wrong. It should be $\Omega_i \cap \Omega_j = \emptyset$. Otherwise, your proof does not hold. I am assuming this was a typo but would like confirmation.

Since there was a bug in the implementation, you *should* at least rerun some of the runs to see if the results are substantially different or not before claiming your results are unaffected.

You should provide some discussion on some closely related work to the scheme presented in Section 3.3. Is it already in use in some practical implementations? How does it relate to the work on multi-task learning such as (but not limited to) "UPDET: UNIVERSAL MULTI-AGENT REINFORCEMENT LEARNING VIA POLICY DECOUPLING WITH TRANSFORMERS" where the action space can vary? How does it relate to the action masking used in SMAC and in "A Closer Look at Invalid Action Masking in Policy Gradient Algorithms"?

# Nice to have (but still important)

MARL doesn't necessarily need to be cooperative or competitive as you mention in the introduction. There is also a lot more than just decentralised/centralised. As mentioned later in the paragraph you can centralise some aspects (training) whilst keeping others decentralised (execution). There is also the large area of communication, in which the agents can communicate with each other at execution and/or training via some channel. I don't think decentralised learning necessarily loses convergence guarantees in all cases (see "An Algorithm for Distributed Reinforcement Learning in Cooperative Multi-Agent Systems" [Lauer and Riedmiller 2000]), and I disagree that this is the reason why CTDE is used.

Missing a reference for parameter sharing from Learning to Communicate with Deep Multi-Agent Reinforcement Learning (Foerster et al. 2016) for introducing it to cooperative deep MARL. Later you say "Agent indication for parameter sharing was first implemented in Foerster et al. (2016),..." but repeatedly claim throughout the previous pages it was first brought to deep rl by "Gupta et al. (2017); Chu & Ye (2017)".

>"... it’s often assumed it can only handle agents with identical behavior in the environment..."

Is this actually an assumption? Surely it follows directly and trivially because all agents are using the exact same policy?

>"This assumption has been relaxed for different types of agents,.."

Adding an agent id to the observations does not relax the assumption, it is one way of ensuring that each agent's policy can be distinct.

The paper is missing references to the mean field literature where all agents are assumed to behave identically.

>"It’s also worth noting that parameter sharing with agent indication is equivalent to naive self-play in competitive environments."

This should be expanded upon to be made a lot more clear. Furthermore, is this even correct? I'm assuming you're referring to naive self-play in which the agent plays with its policy from the previous iteration. How does the agent indicator factor into this at all?

> "...this allows for picking agent indication methods that perform the best overall, and not just with specific (likely non-optimal) hyperparameters."

Do you mean finding the method whose max performance (across hyper-parameters in a specific environment) is the best? The use of overall is a little confusing, especially since you're not even looking at the aggregated performance of the methods across environments.

>"...for two the most popular"

Typo, and should two be here at all? It seems only a single method (PPO) was used.


**Strengths And Weaknesses:**

# Strengths

Section 3.3 presents a simple (in the good sense) and easy to use scheme for allowing parameter sharing when the agents have different action spaces, which could be of large interest to practitioners.

Investigating different schemes for agent indication would be of great interest to many researchers and practitioners in this field.

Overall, the paper is nicely written and easy to follow.

# Weaknesses

## Theoretical results
The headline contributions of the paper, the theoretical results in Section 3, are not substantial or novel enough to be claimed as a contribution.

>"Parameter sharing has been traditionally seen as a technique that can only be used in games with homogeneous agents because of the fact that a single neural network is learned and shared among all agents."

I strongly disagree with this. The references you point to (Foerster et al. 2016 in particular) are keenly aware you require some sort of agent indicator (such as a one-hot encoding of the agent's id) to ensure different policies can be learnt per agent.

## Experimental results
The presentation and analysis of the results in the experimental section could be substantially improved to better fit with the rest of the paper.

Why do the experimental results jump to graphical observations? What about non-image based observation spaces? What type of indicator to add is as much of a question there as it is for image-based observations.

This paper also seems to ignore a very important aspect of parameter sharing, and that is its reasons for being used. Why are implementations even using parameter sharing in the first place? This should be discussed in some detail to provide some motivation for the investigation and to inform the choice and presentation of experimental results. I would also expect a baseline not using any form of parameter sharing to be included as a point of comparison.

Additionally, in the experimental section I am unclear on what you are actually testing. Are you looking to compare the relative performance of the different suggested methods? If so then you should provide graphs/tables comparing the best found hyper-parameters for *each* method.

The paragraph just before Section 4.3 is important in describing your motivations for picking your specific environments and should be investigated a *lot* more thoroughly. You need to include a baseline that features no parameter sharing at all, and then provide the learning curves for all of the different methods. Then some questions that you touch upon but don't discuss are:
- Is there a difference in the asymptotic performance of the methods?
- Do they perhaps end up learning qualitatively different behaviour?

Also, why have these specific methods of agent indication been chosen? They seem quite ad-hoc. Why not try adding a learnt transformation of the agent-id to the observation before feeding it into the network or something inspired by the positional encodings used in transformers? Since you are modifying the architecture's used for each method, why not investigate a method for indication that does not modify the graphical observation directly? You could also concatenate a one-hot vector to a fully-connected layer deeper in the network, etc. There are a myriad of ways to do agent indication, so at the very least some justification should be provided as to why these specifically were chosen.

---

> ### Author Response · Authors · 2022-05-17
> **Response**
>
> First of all, we apologize for the delayed reply, the first author got Lyme disease.
>
> Thank you very much for your careful and attentive review. You raise various concerns about our experimental section, as well as minor writing ones, all of which I believe we address below. You also pointed out a typo in lemma 1; your proposed fix was what it should've been and we've updated it in the new version of the paper, thank you for that.
>
> Regarding the experimental portion of our work, after reading the comments you and the other reviewers made we took another look at that section of the paper, and it was clear that our explanation of the claims we were trying to make in the experimental section were very unclear, in some cases inexcusably so, and there were also a few errors (including the comment about bugs in the conclusion, which should’ve been removed). We’re uploaded a new version of this paper that should _greatly_ clarify the claims we’re making, and it should address all your comments except for a few that we don’t think are applicable in the context of what we’re specifically trying to claim (which may not have been clear before, for which we apologize). If we missed anything else or have questions please let us know and we’ll fix it.
>
> We believe your main concern however was regarding the significance of our proofs. Per your own comments, our paper introduces proofs for generally believed results in multi-agent reinforcement learning which you are unaware of being previously shown, and after a literature review we are similarly unaware of any past works formally showing these results. Aside from the typo you pointed out, you appear to agree our proofs are correct, and your entire criticism is that they are not significant enough. We completely agree with you that our theoretical results are not extraordinarily profound. However, TMLR’s editorial guidelines specifically invite even such modest contributions, and appear to suggest that the lack of significance of the results should not be a reason for rejection (https://www.jmlr.org/tmlr/editorial-policies.html)
>
> We feel that providing the first written rigorous proof of a result — even one that is already widely accepted — should be of some interest to at least some members of the TMLR community.
>
>
> “Nice to Have” comments:
> Everything in paragraphs 1 and 2 of your “nice to have” section should be fixed in the new version, thank you for that (genuinely).
>
> "... it’s often assumed it can only handle agents with identical behavior in the environment..." Is this actually an assumption? Surely it follows directly and trivially because all agents are using the exact same policy?
>
> -> Like our paper showed, if the agent is indicated in the observation, the policy can act differently for each agent by knowing which it is. Without this, this of course isn’t the case, but it’s why we used “assumed” in that sentence.
>
> ‘“This assumption has been relaxed for different types of agents’ … Adding an agent id to the observations does not relax the assumption, it is one way of ensuring that each agent's policy can be distinct.” -> If each agent (or each type of agent) has a distinct policy (that exists within a larger policy) like you said, then parameter sharing is inherently able to learn policies that are capable of controlling heterogeneous types of agents.
>
> “The paper is missing references to the mean field literature where all agents are assumed to behave identically.” -> We added a citation to the original paper for it in MARL, this was apparently accidentally removed during editing.
>
> “‘It’s also worth noting that parameter sharing with agent indication is equivalent to naive self-play in competitive environments.’ … This should be expanded upon to be made a lot more clear. Furthermore, is this even correct? I'm assuming you're referring to naive self-play in which the agent plays with its policy from the previous iteration. How does the agent indicator factor into this at all?” -> I showed this comment to a few colleagues and I’m slightly confused by the issue here if you’d be willing to clarify for us. If, in a competitive game like 2 player Atari Pong, you have a single policy play against itself and update based on the outcomes, this is self play. This is also the exact same update rule used in parameter sharing.
>
> "...this allows for picking agent indication methods that perform the best overall, and not just with specific (likely non-optimal) hyperparameters." … Do you mean finding the method whose max performance (across hyper-parameters in a specific environment) is the best? The use of overall is a little confusing, especially since you're not even looking at the aggregated performance of the methods across environments. -> Your reading was correct, we clarified this sentence in the updated version.
>
> '"...for two the most popular' Typo, and should two be here at all? It seems only a single method (PPO) was used." -> Yes, this is clarified in the new version

---

> > ### Comment · Reviewer_s7mr · 2022-05-23
> > **Replies to some comments**
> >
> > I haven't had a chance to look through the updated manuscript yet, but I thought it best to reply to some of your comments regardless.
> >
> > Regarding the significance of the proofs. The guidelines ask: "Would at least some individuals in TMLR's audience be interested in knowing the findings of this paper?". In relation to the theoretical results, I don't think there are researchers/practitioners in this field that would actually benefit from this formalisation of using agent indicators. They are quite widely used in practise, and I do not think there is any confusion or doubt about whether they theoretically enable different agents to learn distinct policies.
> >
> > About the self-play comment. The choice to use agent indicators in self-play is by no means required, so I do not think the statement "parameter sharing with agent indication is *equivalent* to naive self-play in competitive environments." is true (emphasis on equivalent). You could do self-play in Chess without any kind of agent indication (although it is probably very helpful to do so). For instance the observations to each agent could be the previous and current board, which can tell it all it needs to know about what pieces can be moved. In that scenario the agent's observation spaces are distinct without any agent indication being added. For the example you gave of Atari pong, you could use some kind of agent-centric observation where the controllable agent is always on the left side for instance. In that scenario the observation spaces are no longer disjoint and so both agents will take exactly the same action if they find themselves in the same situation, which is fine and perhaps even expected for naive self-play. Alternatively, you could also just use the exact same observations for both without any agent indication, and hope that a policy with memory can figure out which agent it is controlling (I don't think this is a great setup, but it still very much counts as self-play).

---

> > > ### Author Response · Authors · 2022-06-02
> > > **Response**
> > >
> > > Hey, thank you for your reply, I apologize for the delay on our end again.
> > >
> > > You're probably right that our theoretical results will at least not be of immediate use to practitioners in this space. However, we feel that some of them would still be _interested_ to know that what they're doing has a simple theoretical formalism, derivation and proof of representation power. In the general case, if I were using a method that was popularly used without a derivation/formalism and someone told me it now existed and was simple, my response would likely be "hey that's cool." That's some level of interesting to some number of people, and at least from my understanding of their media statements TMLR was explicitly put together to solicit very odd factually correct documents that would have a very very hard time getting into other ML venues.
> > >
> > > For whatever good it does, I also personally feel that our theoretical work does offer a bit of actual in certain contexts, e.g. as a perhaps a page in textbook on MARL theory that discusses agent indication and for potential future theoretical work on developing better agent indication schemes via statistical learning theory approaches (which to my knowledge has never been previously attempted).
> > >
> > > Also, your comment about self play was absolutely correct. I checked, and this was already fixed in the new version of the paper that we uploaded a bit ago.

---

### Review · Reviewer_Xegx · 2022-05-03

**Summary Of Contributions:**

This work studies different agent indication schemes in the context of parameter sharing for multi-agent reinforcement learning (MARL). A framework for extending agent indication to heterogeneous observation and action spaces is introduced, based on the idea of padding. Next to a formal analysis, an empirical evaluation across five different environments is provided to test the learning capacity and convergence of the proposed approaches.

**Broader Impact Concerns:**

/

**Requested Changes:**

Clarity:

The clarity and readability require a lot of additional effort. In the introduction, for example, the paragraphs do not flow in a coherent manner. There is a jump from discussing CTDE to parameter sharing, without any connection in-between. I would suggest to extend the introduction and improve the flow of the presented ideas. A clear description of the targeted problem setting (e.g., partial observability, cooperative, heterogeneous, etc.), as well as a clear description of the contributions (which are now sprinkled through the whole section) should also be included. More context on parameter sharing, the issues is tackles (e.g., non-stationarity) can also improve the flow of the work.

Section 2 Background and Related work only includes a definition of the mathematical framework (POSGs) and a short description of parameter sharing. POSGs have been present in the literature since before 2017, so I urge to provide a more suitable reference for the model (e.g., Kuhn's (1953) Extensive games and the problem of information). Please incorporate more information on the learning approach used next to the proposed agent indication schemes, as well as a proper related work section (that also identifies similar methods, that should be used as comparison in the experimental section). I would draw attention to the fact that numerous state-of-the-art approaches in deep MARL now include the parameter sharing idea, especially in the CTDE paradigm, so an overview of such methods, together with how the current work builds beyond that would be useful. Augmenting existing approaches with the proposed agent indication schemes would also be interesting to investigate, for example.
Additionally, Definition 1 should be re-checked, it contains mismatched elements in the description, versus the tuple (e.g., U).

Incomplete or unclear statements:
- introduction: "given the apparent usefulness of parameter sharing", the usefulness of parameter sharing was never discussed in the paper
- In the experimental methodology (Section 4.1) - what are the two most popular parameter sharing methods? Only PPO is mentioned.
- "Each environment had standard preprocessing applied to the observations via SuperSuit" - please provide more details on what this entails
- "Agent indication for parameter sharing (...) has been used in numerous derivative works" - such as?
- "We are aware of no work studying the best method of this in graphical observations, theoretically studying methods for this, or attempting to cope with heterogeneous action or observation spaces outside of padding the action space of medivac in Samvelyan et al. (2019)" - More explanation is required here on what 'medivac' is, how padding was done in that case and how the current work is different.

Significance:

Regarding the theoretical results, I fail to see the significance, especially with respect to convergence or learning ability. Furthermore, section 3.3 is more a formal presentation of the implementation of agent indication in heterogeneous settings, which, as mentioned in this work, was already done before.

The empirical results present the top 10 agent indication methods, with only the average reward, without any other comparison to related methods, or baselines. It is also not clear which RL method is used, though I assume PPO. Recent work investigated PPO as a potent baseline in MARL settings (https://arxiv.org/pdf/2103.01955.pdf), so perhaps this could be a fitting baseline here. It is impossible to understand and evaluate the performance of the schemes in these circumstances.

The discussion of the results is also very poor and short, with no insights in the schemes and only shallow remarks such as "arguably surprising result that environments appear to be highly sensitive to agent indication method", "results to not contradict our previous proofs, they just show that the specific method of agent indication can make converging to optimal policies more difficult, like any hyperparameter in machine learning can do". The empirical evaluation requires thus more effort, with a clear set of methods to compare against, a presentation of learning curves, an in-depth analysis of the agent indication schemes and padding methods, in terms of learning speed, convergence, computational complexity.

Finally, it is also alarming to read that there was an issue identified and not mitigated in the experimental setup for some of the used environments. Only stating that "We have no reason to believe that this bug would significantly affect our results, but we cannot state this with complete certainty" is not sufficient in my opinion.

All in all, I find that the claims are not clearly defined and the work also lacks sufficient evidence to support how different agent indication schemes can improve learning in POSGs. This work requires major efforts to improve in terms of writing, scope and contribution clarification and empirical setting and evaluation.

Other remarks:

- citation style 'citep' versus 'citet' is incorrectly used
- I advise to remove the contractions such as it's, can't, etc.
- instead of just the section number, I suggest to refer to sections as `Section #number'.

**Strengths And Weaknesses:**

Parameter sharing is a vital part of many of the state-of-the-art (deep) MARL techniques today, so novel insights in this direction can potentially impact a broad range of existing and future methods. This being said, the scope of this work is unclear. It seems to study agent indication schemes in isolation, with no comparison with related methods and little insights in the results to support any claims on learning capacity or convergence.

Strengths:
- 5 agent indication schemes are tested in 5 environments
- agent indication schemes are extended to heterogenous action and observation spaces, using the idea of padding

Weaknesses:
- weak placement in the literature
- unclear scope and targeted setting
- weak empirical evaluation, with no evidence to support learning capacity of convergence of the proposed schemes
- lack of comparison with other similar techniques (since the scope is unclear, it is also difficult to propose adequate methods here)

---

> ### Author Response · Authors · 2022-05-17
> **Response**
>
> First of all, we apologize for the delayed reply, the first author got Lyme disease.
>
> Given your and the other reviewers comments, we took a very large clarity pass at the work. We addressed all of your comments on the introduction and background related/work specifically and grammatical comments you mentioned elsewhere, as well as all the comments under "incomplete or unclear statements." The only thing we didn't do was to cover new multi-agent RL methods that incorporate partial parameter sharing as we don't fully see the connection to the claims of our work, but if we're missing something with this please let us know. Also, you were correct about the error in definition 1. If you have any additional clarity concerns, please let us know and we’ll try to address them.
>
> The largest change we made regarding the clarity of this work was to the claims and analysis of the experimental sections, which in portions were extraordinarily unclear. Using the same experiments as before, we've extensively updated the discussion in the experimental section (and other sections that discuss the results), and I believe that this should address the vast majority of your comments or should make immediately obvious that they aren't required to substantiate the specific claims our work is trying to make. If there are any additional clarity of detail questions that remain, please let us know and we'll address them. To specifically address the concern about the bug comment in the conclusion, that was included in error from an old version.
>
> Regarding your concerns about the significance of the theoretical results, we generally agree that they are not incredibly significant. However, TMLR’s editorial guidelines specifically invite even very modest contributions, and suggest that the lack of significance of the results should not be a reason for rejection as long as they are all claims are correctly substantiated (https://www.jmlr.org/tmlr/editorial-policies.html).

---

### Review · Reviewer_AW9k · 2022-05-05

**Summary Of Contributions:**

This paper provides the theoretical analysis of parameter sharing techniques for multi-agent reinforcement learning, especially for heterogeneous MASs. Firstly, the authors argue that agent indication is a common parameter sharing technique, however, whether this technique will converge to the optimal policy is unknown. Second, heterogeneous agents with different action spaces cannot use previous parameter sharing techniques, so this paper proposed two padding methods. Third, how to design agent indication for the pixel-based state (observation) is missing and this paper provides several ways. Experiments are conducted on several visual games in the PetttingZoo library.

**Broader Impact Concerns:**

No concerns.

**Requested Changes:**

This paper requires addressing the clarity, a careful proofreading pass, and extensive experiments. Please see the weaknesses part above.

**Strengths And Weaknesses:**

Strengths:

This paper stands from two new perspectives:
1) to analyze the parameter sharing techniques theoretically;
2) design agent indication for pixel-level states (observations).
From these angles, this paper looks interesting and novel. However, there are some technical details that need to be clarified. Experiments are not extensive and convincing.

Weaknesses:

The theoretical analysis is not convincing to me, some details should be specified or clarified.
1) In Sec 3.1, why use $\Omega^{-1}(\omega)$ to denote an agent $i$? $\Omega$ first denotes the joint observation space, it is weird to use $\Omega^{-1}(\omega)$ as a specific agent. It seems just to derive the final equation simply but it does not make sense.
2) The Corollary 1, as well as the Theorem 1 are intuitive and trivial, they do not contain many mathematical details, weakening the contribution of this paper.
3) ‘Two forms of padding’ should be specified clearly. In the introduction section, the authors mentioned they proposed two padding based methods. However, it seems just to use 0-padding for both the observations and actions, which are commonly used in many methods. Maybe I misunderstood this, the ‘two forms of padding’ does not mean the 0-padding for both the observations and actions. As in Sec 3.3, the authors proposed two paddings for actions, one is the joint action space output, and the other is normal 0-padding. However, the experiments do not contain the comparison of the two padding methods.
4) Are there any comparison baselines existing?
5) The experiments are not convincing since there are only tables, and the results do not contain the standard errors.

Based on the above comments, I think this paper requires addressing the clarity, a careful proofreading pass, and extensive experiments, making it unsuitable to TMLR in its current form.

---

> ### Author Response · Authors · 2022-05-17
> **Response**
>
> Thank you so much for your review, we apologize for the delayed reply, the first author got Lyme disease.
>
> To address your weaknesses:
>
> We did a very large clarity pass at the paper based on some other reviewers comments, largely targeting the experimental section and the specific claims we seek to make in it.
>
> 1) We appreciate this feedback and changed this notation to be less confusing in the updated version of the paper. The function simply takes as input an observation and outputs the agent that the observation came from (the function is well defined, since this agent is unique due to the disjointness of the observation spaces).
> 2) The proofs and results are by no means deep or profound, and indeed the agent indication technique has been used in practice. However, we believe there is some value in formally describing this process in terms of POSGs. Additionally, TMLR’s editorial guidelines specifically invite even very modest contributions, and appear to suggest that the lack of significance of the results should not be a reason for rejection as long as claims are correctly substantiated (https://www.jmlr.org/tmlr/editorial-policies.html)
> 3) Your initial reading was correct. The experiments only used the two zero padding methods for knights archers zombies to handle the action and observation space heterogeneity and to show that they experimentally allow for learning, they didn't make any claims about them beyond this.
> 4) We're unaware of any baseline comparisons relevant to the claims we made, could you please clarify what you're requesting just a bit?
> 5) We added a non-table figure that more easily captures the experimental results at a glance. We no longer have access to the data to compute the standard error measurements or we would, but we can't envision a way in which any remotely plausible value of them could make our figures or claims inaccurate or misleading.

---

### Author Response · Authors · 2022-05-17
**Note on New Version**

In the immediate future we'll upload a new version of the paper and reply to all of the reviewers comments. To clarify a bit, portions of the experimental section wound up inexcusably unclear to the reviewers, as well as to the authors upon re-reading the section after a month (despite seeming entirely reasonable to the authors at the time of submission). We largely rewrote the experimental section and the discussion of it in other portions of the paper and had multiple colleagues proof read the paper again and addressed all of the reviewers comments on it where applicable, albeit using the same experiments or making no new claims.

---

### Decision · Action_Editors · 2022-06-13

**Recommendation:** Reject

**Comment:**

The authors have substantially improved their work, based on the reviewers comments. I appreciate the clear responses from the authors, and the sincerity. Each choice has been made with clear intent.

The experimental results, however, do look like they need a bit more work. It is very unfortunate that the data has been lost, making it difficult to show error bars, show performance for each method, and in general provide a more in-depth analysis. Much more could be shown from the extensive experiments that were run. For a resubmission, the key thing is to do this analysis in more depth. It is not necessary to do other environments, or non-image based settings.

The reviewers gave useful suggestions about citing existing literature that uses some of the ideas in this work, such as those for padding and for indicators. A careful exploration of even existing ideas is absolutely worthwhile. It might even be more suitable to say that you explore existing or very natural ideas (rather than that you introduce ideas in padding for example), and that the goal is to provide insight (rather than propose new things).

Finally, a common theme amongst the reviews and the authors response was a disagreement about the significance of the theoretical results. I believe formalizing something that is well-known is worthwhile, even just so that it can be referenced by others. Further, for those new to a field, it may not be so obvious, and an explicit description of even an obvious result is helpful. This work is not being rejected due to a lack of significance in these results.

However, the authors should reconsider whether they want to frame the theory as one of the main contributions of this work. Instead, the focus can be on bringing clarity to this topic, particularly through the empirical study but also by clearly formalizing concepts that are widely believed. Then the formalization is a smaller part. It might also be better to call it a formalization, rather than theory, since I think everyone agrees (including the authors) that the result itself is straightforward. You are absolutely right that TMLR accepts work that is useful or interesting to someone, rather than flashy; this also means that the work can be pitched more honestly, where the experiments are the most significant new contributions with the formalization an additional contribution.

The goals in this work are worthwhile, and the authors have clear intent and sincerity in the work. I would be more than happy to re-review this work as a resubmission.